# UNSUPERVISED OBJECT INTERACTION LEARNING WITH COUNTERFACTUAL DYNAMICS MODELS

**Jongwook Choi**[*,1]   **Sungtae Lee**[*,2]   **Xinyu Wang**[1]   **Sungryull Sohn**[3]   **Honglak Lee**[1,3]
[1]University of Michigan    [2]Individual Researcher    [3]LG AI Research

## ABSTRACT

We present COIL (Counterfactual Object Interaction Learning), a novel way of learning skills of object interactions on entity-centric environments. The goal is to learn primitive behaviors that can control objects and induce their interactions without external reward or supervision being used. Existing skill discovery methods are limited to locomotion, simple navigation tasks, or single-object manipulation tasks, mostly not inducing useful behaviors of inducing interaction between objects. Unlike a monolithic representation usually used in prior skill learning methods, we propose to use a structured goal representation that can query and scope which objects to interact with, which can serve a basis for solving more complex downstream tasks. We design a novel counterfactual intrinsic reward through an use of either forward model or successor features that can learn an interaction skill between a pair of objects given as a goal. Through experiments on continuous control environments such as Magnetic Block and 2.5-D Stacking Box, we demonstrate that an agent can learn object interaction behaviors (e.g., attaching or stacking one block to another) without any external rewards or domain-specific knowledge.

## 1   INTRODUCTION

Reinforcement learning (RL) has achieved remarkable progress at many application domains such as playing games (Mnih et al., 2013; Vinyals et al., 2019), and robotics control (OpenAI, 2018), etc. Very often RL agents are trained to specific tasks, with access to task-specific *extrinsic* rewards. A major drawback of task-specific training is that a proper reward function needs to be given, designed, and tuned so as to achieve desired behaviors, which can be often time-consuming and limits scalability in practice. It is important to be able to solve the task with a very sparse reward signal upon completion/failure of the task, or even without any external task rewards. Unsupervised RL such as task-agnostic exploration or pre-training of skills, aiming at learning interesting or useful behaviors without the use of task rewards or offline data, can provide better initialization or useful macro-actions (skills or options) for building a hierarchical agent to solve more complex and difficult tasks. Unsupervised learning often enables faster learning and achieves better generalization performance when multiple tasks are given after the skill acquiral or pre-training phase.

Despite a number of successes in unsupervised skill discovery (Eysenbach et al., 2018; Sharma et al., 2019; Park et al., 2022) or task-agnostic exploration based on state-entropy maximization or diversity (Pathak et al., 2017; Burda et al., 2019), relatively only a few attempts have been made on environments and tasks with *multiple* objects (e.g. robotics manipulation). In the context of robotics manipulation or (discrete) entity-centric environments other than locomotion or maze navigation environments, exploration can be quite challenging because of this nature of multiple entities. One limitation of novelty-seeking exploration methods in the reward-free context is that exploration would easily converge to a low-hanging fruit behavior where exploration mostly focuses on one particular entity. For instance, in robotics manipulation environments, diversification or novelty seeking of the entire state can be easily dominated by that of the embodied agent itself (i.e., proprioceptive states) or some easy-to-control objects only, as observed and reported in (Zhao et al., 2021; Gu et al., 2021; Park et al., 2022). More interesting primitive behaviors would be interactions between many objects, for more realistic and challenging multi-object tasks such as block stacking (Lee et al., 2021; Sancaktar et al., 2022) or furniture assembly (Lee et al., 2019; Ghasemipour et al., 2022). Notably, some

---

[*]Equal contributions

recent works including (Sancaktar et al., 2022; Cho et al., 2022) present reward-free exploration and skill learning in multi-object manipulation tasks.

In this work, we focus on learning a set of primitive skills that enable interaction between different objects in a task-agnostic, unsupervised fashion. Roughly speaking, interaction between two objects can be described as an action or event that occurs when two objects have a (mutual) effect on each other. Our work leverages an inductive bias that an interaction between objects learned in a task-agnostic manner can be a useful event and hence a useful primitive behavior for solving downstream tasks. Such object-object interactions (as well as agent-object interactions) are usually sparse and difficult to reach with naive exploration, but at the same time can be useful bottleneck states an agent would want to explore and visit often to achieve bigger tasks. In the kitchen, for instance, an interaction between a knife and various ingredients by slicing them with a knife can be one of the basic steps necessary for cooking; when assembling smaller building blocks to build a complex object like furniture, car, or electronic device, connecting matching pieces to form a composite body would be another type of indispensable interaction. As such, it will be important to learn *skills* or primitive behaviors that would induce object-object interactions, in the promise that a hierarchical control that acts upon the interaction skills (Zhang et al., 2021) or chaining of skills in sequence (Slivinski et al., 2020) should solve complex tasks much faster than flat RL agents.

We study how to learn object interaction skills in a very challenging, online, reward-free setting while minimizing the use of domain and task-specific knowledge or task-agnostic offline data, which can be difficult to obtain. More specifically, we learn a goal-conditioned policy where a goal denotes an interaction of which objects is to be made. We derive a novel intrinsic reward that is computed by counterfactual reasoning on the dynamics model (forward models and successor features). The concept of counterfactual reasoning, i.e., "*what if...?*" — predicting or inferring the outcome if something had happened differently (Mesnard et al., 2020; Gajcin & Dusparic, 2022) — naturally aligns with an intuitive interpretation of interaction: interaction is when an object's future state would have been different if it were not for the presence of the other object. We show that the intrinsic reward derived by counterfactual reasoning on object states can efficiently induce the interaction of objects and endow an RL agent with such interaction skills.

Our contribution can be summarized as follows:

- We study a setting of representing goals in terms of entities and objects to interact with, in the context of skill-based RL or goal-conditioned RL.
- We present a novel intrinsic reward algorithm COIL (Counterfactual Object Interaction Learning), which is able to learn a policy that makes the goal objects interact with each other, in a reward-free unsupervised exploration setting.
- We show that such an entity-centric interaction skill is generalizable to unseen, more object setting.

## 2 APPROACH

### 2.1 PRELIMINARIES AND NOTATIONS

Throughout the paper, we consider the task as an MDP $\mathcal{M} = (\mathcal{S}, \mathcal{A}, \mathcal{P}, \mathcal{R}, \gamma)$, where $\mathcal{S}$ is a state space, $\mathcal{A}$ is an action space, $\mathcal{P}$ is a transition probability, $\mathcal{R}$ is a (extrinsic) task reward function, and $0 \leq \gamma < 1$ is a discount factor. Our goal is task-agnostic, unsupervised skill learning with no extrinsic rewards. We assume that the state space $\mathcal{S}$ can be explicitly factorized as the Cartesian product $(\mathcal{S}_{\text{object}})^N \times \mathcal{S}_{\text{agent}}$ where $N$ is the number of objects. We also assume the object space is permutation-invariant, i.e., $\{o_1, \cdots, o_N\} \in \mathcal{S}_{\text{object}}$ is a *set*. Such a structural representation is common in robotics control (Keramati et al., 2018; Zhao et al., 2021; Sancaktar et al., 2022) and is a mild assumption. However, our method is not necessarily limited to state-based control only, as one can combine with existing entity-centric representation learning methods from pixel observations (Watters et al., 2017; Greff et al., 2019; Xu et al., 2019; Veerapaneni et al., 2020; Locatello et al., 2020).

**Goal representation.** Skills are usually modeled in the form of goal-conditioned policies, $\pi(a|s,g)$ where $g \in \mathcal{G}$ represents a *goal*. Common choices for goal $g$ include full state observation, a hand-crafted goal with domain knowledge, or latent variables. Our particular choice is a pair of objects, "*target*" and "*affector*", i.e., $g = (o^T, o^A) \in [N] \times [N]$. A semantic meaning for this goal representation would be that two objects $o^T$ and $o^A$ should have an interaction, such that the **affector**

Figure 1: A high-level overview of COIL. Suppose an interaction was made, then a counterfactual intervention on the affector object $o^A$ (e.g., putting it aside or change the object state randomly) would have made the future state of the target object $o^T$ different. If no interaction was made, the target would remain in the same state regardless of the counterfactual intervention. We measure the discrepancy of the target object with and without the counterfactual intervention, which becomes the intrinsic reward for interaction.

$o^A$ affects the **target** $o^T$ as a result of the interaction. In our settings, for the sake of simplicity, we assume the reference to objects $o^T, o^A$ are simply categorical indices (or pointers), namely $T, A \in [N] = \{0, 1, \cdots, N - 1\}$, respectively. However, more in general (e.g., for image observations), the goal representation for target objects can be replaced with a continuous vector to represent a reference to an *arbitrary* object in the current state, which we leave as a future work.

## 2.2 Learning Interaction Skills with Counterfactual Forward Model

How can we learn interaction skills for two given objects, and how can we learn a reward function that would incentivize interactions between two objects? Our goal is to simultaneously learn such a reward function and agent's object-object interaction skills in a *reward-free* setting.

Our main idea is to use a *counterfactual reasoning*; i.e., predict what would have happened instead, if other objects involved in an interaction were not there or were in a different state. This form of inductive bias can provide us with a useful learning signal for interaction learning without relying on an external task reward.

Given a trajectory of observations as object states, we want to identify whether an interaction between two objects happened or not. When a (physical) interaction between two objects happened, these two objects would have affected each other. In other words, the future state of an object would have been different without a specific configuration of the other object, provided that interaction happened. On the contrary, when there was no interaction between them the future state of an object would remain the same or not dramatically different regardless of the other object. A motivating example is depicted in fig. 1.

We can formalize this idea as follows. Consider a MDP transition observed by an agent, $(s_t, a_t, s_{t+1})$ where $s_t = \{o_t^T, o_t^A, \cdots\}$ and $s_{t+1} = \{o_{t+1}^T, o_{t+1}^A, \cdots\}$ (without the loss of generality) for a pair of objects $o^T$ and $o^A$ given as a goal $g$. Suppose an interaction between the target and the affector happened, where the target got affected by the affector in the interaction. Then, if we made an *counterfactual intervention* on the affector object $o^A$, i.e., changing the object state $o_t^A$ randomly with $\widetilde{o_t^A}$ to obtain an *intervened state* $\widetilde{s_t} = \{o_t^T, \widetilde{o_t^A}, \cdots\}$, the same action $a_t$ applied on $\widetilde{s_t}$ would have resulted in a different next state $\widetilde{o_{t+1}^T}$ of the target object than its actual next state $o_{t+1}^T$. In other words, the discrepancy between the actual next state $o_{t+1}^T$ and the counterfactual next state $\widetilde{o_{t+1}^T}$ will be high. On the other hand, when there was no interaction happened between the two in this transition, we can expect that $o_{t+1}^T$ would remain the same regardless of the intervention $\widetilde{o^A}$ on the affector, i.e., it would be that $\widetilde{o_{t+1}^T} = o_{t+1}^T$. To put together, $\|o_{t+1}^T - \widetilde{o_{t+1}^T}\|^2$ can quantize the interaction between the target and the affector.

However, the counterfactual next state $\widetilde{s}_{t+1}$ is not actually observed by an agent. So we can instead predict the object $A$'s next state by learning a forward dynamics model:

$$\widetilde{o_{t+1}^T} = f_{\text{forward}}(o_t^T, \widetilde{o_t^A}, a_t, s^t \setminus \{o_t^T, o_t^A\}) \tag{1}$$

This gives us a counterfactual interaction reward function: we first compute an intervention $\widetilde{o_t^A}$ on the affector object, and plug it to the forward model to predict the next state $\widetilde{o_{t+1}^T}$ of the target object. Intervention of the affector can be implemented in many ways, such as random perturbation of the state vector by adding Gaussian noises, but an easy yet effective way of in-distribution randomization is to randomly sample the object vector from the replay buffer.

Finally, we define the counterfactual interaction reward $r_{\text{COIL-Forward}}(s_t, a_t, s_{t+1}) = \|o_{t+1}^T - \widetilde{o_{t+1}^T}\|^2$, which can be maximized by any underlying RL method (e.g., SAC or DQN) with a simultaneous learning of the forward model and the RL policy. We call this resulting algorithm **COIL** (Counterfactual Object Interaction Learning) and specifically this variant of using forward model **COIL-Forward**.

## 2.3 LEARNING INTERACTION SKILLS WITH COUNTERFACTUAL SUCCESSOR FEATURES

In this section, we will present an improvement to COIL-Forward, called **COIL-SF**. One downside of the above COIL-Forward is that it assumes the counterfactual intervention would have an immediate, easily distinguishable change in the very next time step. In many realistic environments, the effect and consequence of interaction is *delayed* to be discernible enough; the change actually exists in the true world state but an observer would not be able to recognize the subtle difference until a few time step has elapsed. Therefore, it is practically very important to take long-term futures into consideration to correctly evaluate the consequence of a counterfactual intervention.

One natural way to deal with this problem would be to learn a multi-step, recurrent forward dynamics model. However, learning such a forward model can be challenging due to uncertainty and the quick accumulation of prediction errors Moerland et al. (2020); Lutter et al. (2021). Instead of learning a multi-step forward model, we propose to use the successor features (SF) framework (Dayan, 1993; Barreto et al., 2016) to incorporate long-term futures that can still derive a reward signal for interaction learning.

A successor feature $\Psi^\pi(s, a)$ of a state $s$ with respect to a policy $\pi$ is an expected discounted sum of the feature of future states to be visited when starting from the state $s$ and the action $a$, and following the policy $\pi$ thereafter:

$$\Psi^\pi(s, a) = \mathbb{E}_\pi \left[ \sum_{t=0}^{\infty} \gamma^t \Phi(s_t) \;\middle|\; s_0 = s, a_0 = a \right]. \tag{2}$$

where $\Phi(s_t)$ is called the cumulant, which is the feature of future states to accumulate. Successor features can be seen as an instance of generalized value functions (GVF) (Sutton et al., 2011) that *predicts the future* and summarize what will happen in the future for a state $s$ in some specific form, which can be easier than directly predicting the next states. Successor features can be learned using simple TD learning like Q-learning (Dayan, 1993).

To derive a reward function that tells whether an interaction is made or not, let's consider two objects $T$ (target) and $A$ (affector) given at hand and the future of $T$ when an intervention is made on the object $A$. For this, we consider an entity-centric successor feature whose cumulant function is simply the readout $o_t^T$ of the target object in the state representation $s_t$:

$$\Psi_{\text{target}}^\pi(s, a) = \Psi_{\text{target}}^\pi(\{o^T, o^A, \cdots\}, a) = \mathbb{E}_\pi \left[ \sum_{t=0}^{\infty} \gamma^t o_t^T \;\middle|\; s_0 = s, a_0 = a \right] \tag{3}$$

for a query state $s = \{o^T, o^A, \cdots\}$. As in COIL-Forward, let's suppose we make a counterfactual intervention on $B$ at timestep $t$ to get the intervened object state $\widetilde{o^B}$ from $o_B$. Denoting $\widetilde{s} = \{o^T, \widetilde{o^A}, \cdots\}$, the reward function for interaction can be written as

$$r_{\text{COIL-Successor}}(s, a, s') = \|\Psi_{\text{target}}^\pi(s, a) - \Psi_{\text{target}}^\pi(\widetilde{s}, a)\|^2. \tag{4}$$

We call this variant of using successor features for learning interactions **COIL-SF**. When there was no interaction happened between $T$ and $A$, the entity-centric successor features $\Psi_T^\pi$ will be the same regardless of the intervention, in which case $r_{\text{COIL-SF}}$ would be 0. Note that, in practice, rewards for non-interaction transitions might be slightly bigger than 0 due to the epistemic uncertainty of the

model. On the other hand, if the future state of $T$ would have changed much due to the intervention on $A$, the SF values $\Psi_T^\pi(s, a)$ and $\Psi_T^\pi(\widetilde{s}, a)$ will be different, in which $r_{\text{COIL-SF}}$ will evaluate to a higher scalar value. Appendix C presents an analysis of the learned reward function for different types of states.

## 3  RELATED WORK

Our work builds upon three big areas of prior research in reinforcement learning.

**Object-Oriented RL**  Object-oriented RL (Diuk et al., 2008) aims at improving data efficiency and generalization by leveraging representation of multiple objects and their relations. C-SWM (Kipf et al., 2019) proposes a GNN-based network to learn the world model of the object-based task using contrastive learning. Compared to models based on pixel reconstruction, C-SWM provides a rich representation of objects. CEE-US (Sancaktar et al., 2022) utilizes the epistemic uncertainty of structured world model (Kipf et al., 2019) as an intrinsic reward and uses it to gather data for the world model training. The world model is then used for planning to solve downstream tasks. While they also utilize an inductive bias that object-object interaction is useful, they learn such information in the world model while we learn it directly in the policy. Also, they consider agent-object interaction along with object-object interaction while we focus on object-object interaction.

**Exploration**  Cho et al. (2022) proposed a mutual-information (MI) based exploration algorithm to induce interactions between the *agent* and an object, which combines the MUSIC objective (Zhao et al., 2021), i.e., MI between agent and object, and the diversity term similar to DADS (Sharma et al., 2019) for the object's future state. Seitzer et al. (2021) used object-centric causal action-influence as an intrinsic reward. However, interactions between different objects are not considered; the skills are limited to simple control of a single target object specified by the task. Very recently, Sancaktar et al. (2022) proposed curiosity-based exploration algorithm CEE-US that can learn object-object interactions in an unsupervised fashion, with intrinsic reward being the epistemic uncertainty through ensemble disagreement (Pathak et al., 2019). This work is the closest to our work, but despite GNN's ability to generalize to multiple objects during planning, their monolithic skill representation is limited to be useful for hierarchical learning or planning.

Several papers have proposed exploration methods using successor features (SF). Zhang et al. (2019) use the difference of SF between consecutive states as an intrinsic reward to efficiently explore bottleneck states. Machado et al. (2020) propose an inverse of the L1-norm of the SF as a variant of count-based exploration. Hoang et al. (2021) utilize SF to define the distance function between states and learn a goal-conditioned policy to drive exploration. However, to the best of our knowledge, SF has not been used in object-centric environments and has not been combined with counterfactual reasoning.

**Counterfactual Reasoning**  Buesing et al. (2018) use a structural causal model in POMDP, which generates counterfactual trajectories for background planning, leading to a better sample efficiency and smaller bias of the prediction in guided policy search. Sharma & Kroemer (2020) utilize an inductive bias that, in similar scenes, if similar action has been taken it would give similar results. They utilize contrastive learning in object-centric tasks to acquire an object relation model, which is subsequently utilized in real-world precondition learning tasks. Counterfactual Credit Assignment (Mesnard et al., 2020) utilizes counterfactual reasoning on action to achieve unbiased, low variance credit assignment. Most approaches do counterfactual inference on the agent's action, i.e., concerns what would have happened if the agent made a *different decision*; our approach differs in the sense that our counterfactual intervention is made on the object states instead of the agent's action.

## 4  EXPERIMENTS

### 4.1  ENVIRONMENTS

In the experiments, we test our proposed approach on multi-object continuous control environments: a toy environment (**StackingBox**) and more challenging environment (**Mangetic Blocks**).

**Stacking Box.**  Stacking Box is a 2.5-D continuous control environment in which a cursor agent and multiple box-shaped objects of the same size are randomly spread throughout a fixed arena. The

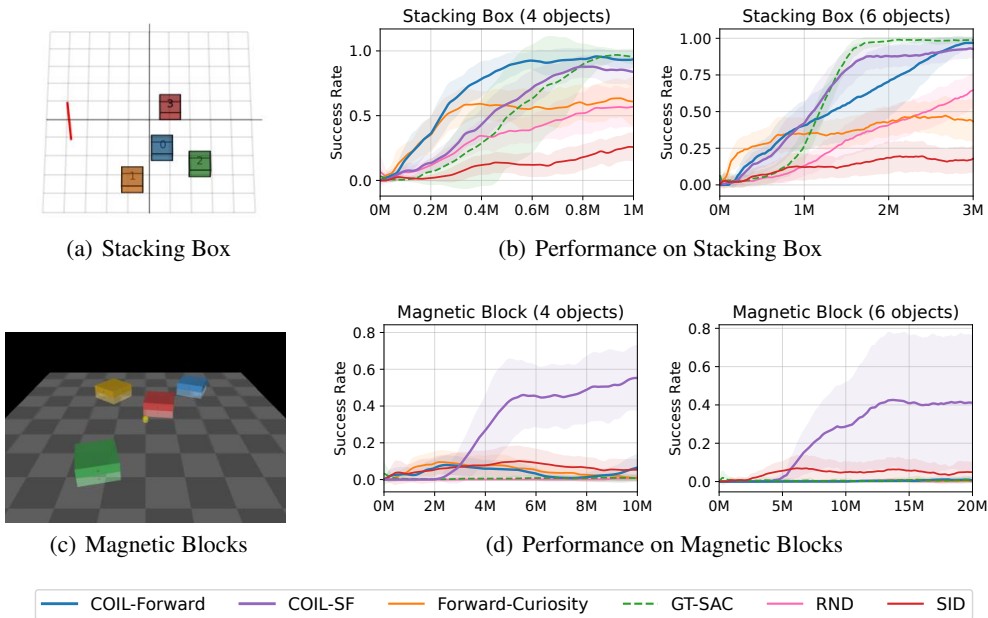

(a) Stacking Box       (b) Performance on Stacking Box

(c) Magnetic Blocks       (d) Performance on Magnetic Blocks

Figure 2: Progress of the success rate on the Stacking Box and the Magnetic Blocks environment. Runs are averaged over 3 random seeds. See Section 4.3 for analyses and interpretation of the result.

agent can move in any direction within the xy plane and can grab an object that overlaps with the agent. If the agent moves towards an object while holding another object, the object being held and moved will be placed on top of any other existing object. We assume that the height of each object is quantized to integer values (such as $0, 1, 2, \ldots$). The process of stacking one object onto another occurs instantly in a single MDP transition.

**Magnetic Blocks.** Magnetic Blocks is a continuous control environment in which an embodied cursor agent can interact with square-shaped block objects. The agent has a continuous action space that includes movement (translation), rotation, and grabbing through control of the joint's torque. The agent can move freely within the arena and can grab an adjacent object by slightly lifting up and moving around the object, or rotating it along with the agent. When the agent moves a held object close enough to another object such that the two objects become parallel, they will be connected by magnetic force. If the edges are not parallel, one object will push the other. A distinctive interaction in this environment is observed when two objects become connected through magnetic forces and then move together in unison.

### 4.2 IMPLEMENTATION DETAILS

Taking the factorized state representation into consideration, we use a network with scaled dot-product attention architecture (Vaswani et al., 2017) to transform object states into desired outputs (actor, critic, and forward/successor models). The full network architecture is shown in Figure 4. We note that by sharing the weights for key and value matrices on objects other than target and affector, the network becomes permutation-invariant over the order of objects other than the goal objects (i.e., $o^T$ and $o^A$). COIL alternatingly updates the policy (actor and critic) and the model (forward model or successor features). For RL algorithm, we use SAC (Haarnoja et al., 2018) although COIL can be combined with any RL algorithms. More details can be found in the appendix.

### 4.3 PERFORMANCE OF LEARNING OBJECT INTERACTION SKILLS: QUANTITATIVE RESULTS

We first study how well the proposed approach (COIL) can learn object interaction skills in a reward-free setting, with a comparison to strong exploration methods. At the beginning of every episode, a goal $g = (T, A)$ is chosen randomly to specify which objects should interact.

(a) smaller size     (b) smaller size     (c) more objects, smaller size     (d) more objects

Figure 3: Progress of the success rate when fine-tuning from a COIL-SF agent *pre-trained on the* 4 objects (object size 100%) setting in Magnetic Blocks environment. Runs are averaged over 3 random seeds. See Section 4.5 for more analyses.

**Baselines.** (1) Sparse-GT: A SAC agent trained to maximize the *sparse* ground-truth interaction reward, where the per-step reward is 1 if a correct interaction between the affector and the target is made (e.g., stacking or magnetic connection) or 0 otherwise, which is the same as the success metric. (2) Forward-Curiosity: this maximizes the prediction error of the forward model for the target object as an intrinsic reward: $\|o_{t+1}^T - f_{\text{forward}}(s_t, a_t, g_t)\|^2$. (3) SID (Zhang et al., 2019): this maximizes the "successor feature control" reward: $\|\Psi(o_{t+1}^T) - \Psi(o_t^T)\|^2$. (4) RND (Burda et al., 2019): this maximizes the prediction error of a randomly initialized network's feature representation of the target object's state as an intrinsic reward: $\|f_{\text{random}}(o_{t+1}^T) - f(o_{t+1}^T)\|^2$.

For object-centric tasks, interactions can lead to significant changes in the object's state, making it desirable to employ curiosity-based exploration methods as baselines. RND is a state-of-the-art exploration method that seeks novel states, and Forward-Curiosity and SID are curiosity-based exploration techniques that use the Forward Model and Successor Feature, respectively.

**Quantitative Results.** The success rate of the algorithms is displayed in Figure 2, based on the evaluation episodes. Successful outcome is defined as the stacking of the target on the affector in Stacking Box and the connection of the two selected goal blocks in Magnetic Blocks.

**Stacking Box.** COIL algorithms converge to a success rate of approximately 1.0, while curiosity-based exploration methods show limitations with upper bounds in their performance. One thing to note is that COIL-Forward outperforms COIL-SF in Stacking Box with 4 objects. In the Stacking Box environment, interactions occur instantaneously, enabling the 1-step forward model of COIL-Forward capture the occurrence of the interaction. This is supported by an analysis of the error of the dynamics model (see Figure 8). Transitions involving interactions exhibit a significantly higher ratio of the counterfactual prediction error (i.e., the prediction error when counterfactual intervention is made) to epistemic uncertainty, compared to transitions without events. On the other hand, Forward-Curiosity, SID, and RND are limited to manipulating individual objects without learning interaction stably (see Figure 6).

**Magnetic Blocks.** COIL-SF is the only algorithm that successfully learns interaction skills between objects. Despite leveraging domain knowledge regarding the occurrence of interactions, Sparse-GT fails to learn even the basic task of grabbing an object (see Figure 7). Forward-Curiosity, SID, and RND can learn how to grab an object but interaction between the objects barely happen. This suggests that learning to induce interactions between objects in Magnetic Blocks is a challenging exploration problem, unlike the Stacking Box environment.

We find COIL-Forward is not effective enough to learn interactions in Magnetic Blocks, which accords with the motivation discussed in 2.3. In this environment, interactions make only a subtle difference in the object's state during a single-step transition and can be better discerned only in longer-term future; we verify this by analyzing the dynamics model errors (see Figure 9). When interactions occur, the counterfacutal prediction error is not significantly higher than the epistemic uncertainty in the forward model (in COIL-Forward). However, the counterfactual prediction error of the successor feature (in COIL-SF), is significantly higher than the epistemic uncertainty despite the counterfactual intervention, so the interaction reward could lead to learning interactions.

### 4.4 QUALITATIVE RESULTS

In Stacking Box, a typical interaction behavior that COIL learns is to stack the target on the affector. Note that a target should be on top of an affector to say interaction happened. If the affector is on top

of the target, perturbing the affector's state does not affect the target's state. An interesting finding was that the agent repeatedly stacked and unstacked the boxes, resulting in multiple interactions within a single episode.

In Magnetic Blocks, the interaction behavior is to grab the target and connect it to the affector by moving and rotating. Note that the agent needs to rotate accurately to connect the blocks, which makes Magnetic Blocks a hard exploration problem in terms of learning to interact.

We present snapshots of COIL-SF's typical behaviors in Magnetic Blocks in Figure 5. Various ways of interaction between the target and the affector after they are connected are observed: (1) the agent grabs the affector and moving the both blocks, causing the magnetic force to pull the target; (2) the agent grabs the target and pushes the connected block towards the affector direction until it reaches the wall, causing the target to be blocked by the affector; and (3) the agent grabs the target and moving both objects together, which can be seen as some kind of interaction, because attaching the affector changes the physical properties of the connected blocks, such as the force required to move them or their resulting velocity. Among the three, COIL-SF tends to learn the second type of interaction, where the agent grabs the target and pushes it toward the affector until it hits the wall.

### 4.5 GENERALIZATION TO MORE/UNSEEN OBJECTS

We evaluate the task-agnostic skills learned by COIL-SF, testing whether they can be applied to environments with more and unseen objects. First, the policy and successor feature networks are pre-trained on Magnetic Blocks with 4 objects for 10 million steps and perform fine-tuning for 1 million additional steps. For each setup, the performance of COIL-SF fine-tuned from pre-trained networks is compared to that of COIL-SF trained from scratch for (10+1) million steps, ensuring a fair comparison. We tested the generalization ability on 4 different setups with varying object sizes and numbers: (a) 4 objects, 33% object size, (b) 4 objects, 66% object size, (c) 6 objects, 100% object size, and (d) 6 objects, 66% object size.

**Unseen objects: (a), (b)**  To test the generalization ability of COIL-SF on unseen objects, we varied the scale (size) of the objects by 33% or 66%. The Figure 3 (a-b) show the performance of COIL-SF fine-tuned on pre-trained networks. When tested on the 66% scale, COIL-SF gets a high success rate even without any training. When tested on the 33% scale, the initial performance of COIL-SF is poor, but the performance improves rapidly within 1 million steps of further training while learning COIL-SF from the scratch fails.

**More and Unseen objects: (c), (d)**  To test the generalization ability of COIL-SF on a different number of objects, we conducted experiments with more objects and varying scales (66%, 100%). The Figure 3 (c-d) show the performance of COIL-SF. Surprisingly COIL-SF fined-tuned on pre-trained networks performs better even in more and unseen objects settings indicating that skills learned from COIL-SF can be used as task-agnostic skills.

Overall, the successful learning of task-agnostic skills with COIL-SF has implications for future research, as these skills could potentially be incorporated into hierarchical reinforcement learning for more complex tasks.

## 5 CONCLUSION

In this paper, we introduce **COIL** (Counterfactual Object Interaction Learning), a novel approach to learning object-object interaction skills using intrinsic rewards, and the concept of counterfactual dynamics. Our results demonstrate that COIL can effectively learn to interact with objects in challenging continuous, object-centric environments outperforming all of the baselines including Sparse-GT, which incorporates task-specific knowledge. We also showed that an entity-centric interaction skill learned by COIL is generalizable to unseen, more object setting. Given the complexity and diversity of real-world tasks such as furniture assembly or learning to cook, we believe that unsupervised learning of object-object interactions is crucial, and COIL represents a significant step towards this goal. Considering that the real world tasks contain multiple modes of interaction and complex state representation, combining diverse skill learning (Eysenbach et al., 2018; Sharma et al., 2019; Park et al., 2022) and object-centric representation learning methods (Locatello et al., 2020) will be an interesting future work.

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

# Appendix

## A  DETAILS OF IMPLEMENTATION AND EXPERIMENTS

### A.1  STACKING BOX

The state of the agent is denoted by its $(x, y)$ coordinates, while each object is represented by $(x, y, z, held)$, where the binary value of *held* indicates whether the object is in the grasp of the agent or not. The action space is three-dimensional and includes the variables $\Delta x$, $\Delta y$, and *grab*. The range of values for each variable is from -1 to 1, where $\Delta x$ and $\Delta y$ denote the displacement of the agent's movement and *grab* indicates whether to make a grab. If the value of *grab* is 0 or greater, the agent will grab the object; otherwise, it will release it.

### A.2  MAGNETIC BLOCKS

The state of the agent and each object are 9-dimensional vectors: $(x, y, z, \cos\theta, \sin\theta, v_x, v_y, v_z, \omega)$ where $\cos\theta$ and $\sin\theta$ represent a 2D Euler rotation, $v$ is the velocity, and $\omega$ is the angular velocity with respect to the joint. The action space is four-dimensional and includes the variables $F_x$, $F_y$, $\tau$, and *grab*. The range of values for each variable is from -1 to 1, where $F_x$ and $F_y$ denote the motor translation torques, $\tau$ the rotation torque, and *grab* indicates whether to make a grab. If the value of *grab* is 0 or greater, the agent will grab the object that overlaps with the agent; otherwise, it will release it.

### A.3  IMPLEMENTATION DETAILS

In Stacking Box and Magnetic Blocks, each episode has a length of 200. The size of the replay buffer is one million and we start updating the policy after 10000 steps to fill the replay buffer. We use soft target update with the ratio of $\tau$. For the forward and successor models, we use Scaled Dot Product Attention Network to get the embedding vector of objects. Then dense layers are used at the end. For RL algorithm, we use SAC (Haarnoja et al., 2018) though COIL can be combined with any RL algorithms. We search over the hyper-parameter range in Table 1 for Stacking Box and Table 2 for Magnetic Blocks. The hyper-parameters that give the highest AUC (area under the curve) in the success rate for each task are chosen as the best hyper-parameters.

| COIL-Forward / Forward Curiosity | | | |
|---|---|---|---|
| **Hyperparameters** | **Sweep range** | **n=4** | **n=6** |
| Actor-Critic network hidden dimensions | [64,64], [256,256] | [256,256] | [256,256] |
| Initial temperature | 0.001, 0.01, 0.1, 0.3, 1 | 0.1 | 0.1 |
| $\tau$ for SAC target update | 0.001, 0.01, 0.1, 0.2, 0.3 | 0.01 | 0.01 / 0.001 |
| Forward model hidden dimensions | [64,64], [256,256] | [256,256] | [256,256] |
| Reward scale | 10, 100, 1000 | 10 | 10 |

| COIL-SF / SID | | | |
|---|---|---|---|
| **Hyperparameters** | **Sweep range** | **n=4** | **n=6** |
| Actor-Critic network hidden dimensions | [64,64], [256,256] | [256,256] | [256,256] |
| Initial temperature | 0.001, 0.01, 0.1, 0.3, 1 | 0.1 | 0.1 |
| $\tau$ for SAC target update | 0.001, 0.01, 0.1, 0.2, 0.3 | 0.01 / 0.001 | 0.001 |
| SF model hidden dimensions | [64,64], [256,256] | [256,256] | [256,256] |
| SF model target update period | 1, 5, 10 | 5 | 5 |
| SF model discount factor | - | 0.8 | 0.8 |
| $\tau$ for SF target update | 0.001, 0.01, 0.1, 0.2, 0.3 | 0.01 / 0.001 | 0.001 |
| Reward scale | 1, 10, 100 | 10 | 10 |

| Sparse GT-SAC / RND | | | |
|---|---|---|---|
| **Hyperparameters** | **Sweep range** | **n=4** | **n=6** |
| Actor-Critic network hidden dimensions | [64,64], [256,256] | [256,256] | [256,256] |
| Initial temperature | 0.001, 0.01, 0.1, 0.3, 1 | 0.1 | 0.1 |
| $\tau$ for SAC target update | 0.001, 0.01, 0.1, 0.2, 0.3 | 0.01 | 0.001 |
| Reward scale | 10, 100, 1000 | 10 / 100 | 10 / 100 |

Table 1: Hyperparameters swept over and the final values used in Stacking Box. $n$ denotes the number of objects.

| COIL-Forward / Forward Curiosity | | | |
|---|---|---|---|
| **Hyperparameters** | **Sweep range** | **n=4** | **n=6** |
| Actor-Critic network hidden dimensions | [64,64,64,64], [256,256] | [256,256] | [256,256] |
| Initial temperature | 0.001, 0.01, 0.1, 0.3, 1 | 0.01 | 0.01 |
| $\tau$ for SAC target update | 0.001, 0.01, 0.1, 0.2, 0.3 | 0.3 | 0.1 / 0.3 |
| Forward model hidden dimensions | [64,64,64,64], [256,256] | [256,256] | [256,256] |
| Reward scale | 10, 100, 1000 | 10 | 10 |

| COIL-SF / SID | | | |
|---|---|---|---|
| **Hyperparameters** | **Sweep range** | **n=4** | **n=6** |
| Actor-Critic network hidden dimensions | [64,64,64,64], [256,256] | [256,256] | [256,256] |
| Initial temperature | 0.001, 0.01, 0.1, 0.3, 1 | 0.01 | 0.3 / 0.01 |
| $\tau$ for SAC target update | 0.001, 0.01, 0.1, 0.2, 0.3 | 0.01 / 0.1 | 0.01 / 0.2 |
| SF model hidden dimensions | [64,64,64,64], [256,256] | [64,64,64,64] | [64,64,64,64] |
| SF model target update period | 1, 5, 10 | 5 | 5 |
| SF model discount factor | - | 0.8 | 0.8 |
| $\tau$ for SF target update | 0.001, 0.01, 0.1, 0.2, 0.3 | 0.01 / 0.1 | 0.01 / 0.2 |
| Reward scale | 1, 10, 100 | 10 | 10 |

| Sparse GT-SAC / RND | | | |
|---|---|---|---|
| **Hyperparameters** | **Sweep range** | **n=4** | **n=6** |
| Actor-Critic network hidden dimensions | [64,64,64,64], [256,256] | [256,256] | [256,256] |
| Initial temperature | 0.001, 0.01, 0.1, 0.3, 1 | 0.01 | 0.01 |
| $\tau$ for SAC target update | 0.001, 0.01, 0.1, 0.2, 0.3 | 0.1 | 0.3 |
| Reward scale | 10, 100, 1000 | 10 / 100 | 10 / 100 |

Table 2: Hyperparameters searched over and the final values in Magnetic Blocks. $n$ denotes the number of objects.

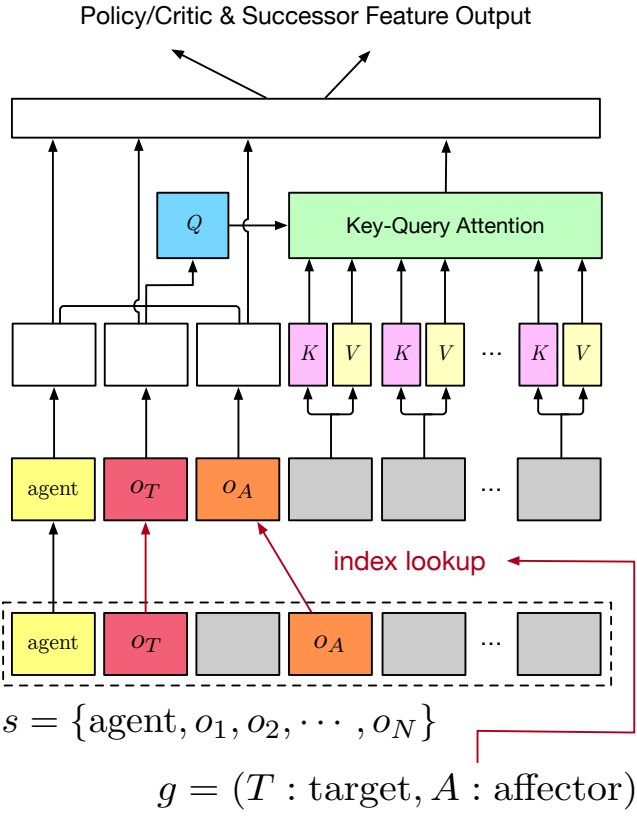

Figure 4: A network architecture used in the experiments.

## B  QUALITATIVE EXAMPLES

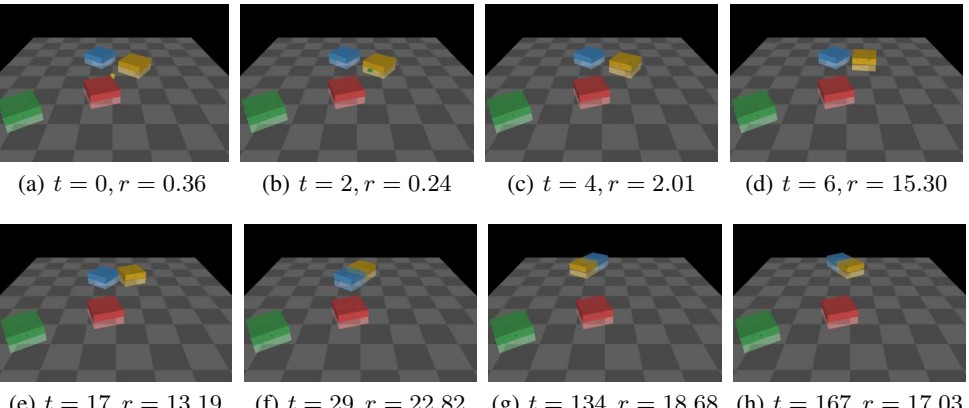

(a) $t = 0, r = 0.36$    (b) $t = 2, r = 0.24$    (c) $t = 4, r = 2.01$    (d) $t = 6, r = 15.30$

(e) $t = 17, r = 13.19$    (f) $t = 29, r = 22.82$    (g) $t = 134, r = 18.68$    (h) $t = 167, r = 17.03$

Figure 5: Snapshots of COIL-SF in Magnetic Blocks. In this episode, the target is the yellow object and the affector is the blue object. (a) Initial state, (b) Agent heads towards the target, (c) Agent grabs the target, (d) Agent heads towards the affector while holding the target, (e) Agent rotates the target to align two objects, (f) Agent connects the target and the affector (**a successful interaction**), (g-h) Agent pushes the affector to somewhere near the wall. The amount of reward $r_{\text{COIL-Successor}}$ the agent receives is shown in the caption of each episode; we can see that the per-step reward is highest when a correct interaction (magnetic connection between the objects in the specified goal) is made.

## C  ANALYSIS OF COIL-SF REWARD

To analyze what reward function COIL-SF has learned, we labeled each state with the following 7 categories on the **Magnetic Blocks** environment with 4 objects.

- Grab-T: the agent is grabbing the target object.
- Grab-A: the agent is grabbing the affector object.
- Connect-TA: the target and affector objects are correctly connected. Note that when target and affector objects are connected, highly likely the target object will be affected by the affector object, i.e., interaction occurs.
- Connect-TA-Only: a subset of Connect-TA states. Note that Connect-TA states include states where objects other than the target and affector also are connected to the target and affector objects. However, Connect-TA-Only excludes such states.
- Connect-TX: the target object is connected to a wrong object (X), i.e., anything but the affector object.
- Connect-AX: the affector object is connected to a wrong object (X), i.e., anything but the target object.
- No-Event: all other states not included in the above 6 categories.

Table 3 shows an average reward given to states with each label, for a successful instance of COIL-SF. Among the 7 labels, Connect-TA-Only receives the highest rewards. Connect-TA receives a slightly lower reward than Connect-TA-Only. Considering that Connect-TX or Connect-AX receive small rewards, we assume that a small portion of Connect-TA states are the states where objects other than the target and effector are also connected, and those states have small rewards. Grab-T and Grab-A receive high rewards compared to No-Event or Connect-TX or Connect-AX. This may be due to Grab-T having an intersection with Connect-TA-Only, which is a set of states where the target and the affector objects are connected and the agent is grabbing the target object.

| State Labels | Average Reward | Relative Ratio |
|---|---|---|
| No-Event | 0.7 | 0.040 |
| Grab-T | 14.21 | 0.803 |
| Grab-A | 10.7 | 0.604 |
| Connect-TA | 17.3 | 0.977 |
| **Connect-TA-Only** | **17.7** | **1.0** |
| Connect-TX | 0.51 | 0.029 |
| Connect-AX | 1.8 | 0.101 |

Table 3: Average COIL-SF reward given to the 7 types of states on the **Magnetic Blocks** environment. COIL-SF gives the highest reward to Connect-TA-Only.

## D  ADDITIONAL PLOTS

We provide additional plots for further analysis of COIL algorithms. Figure 6 and Figure 7 support the quantitative result that COIL learns to make interactions while curiosity-based methods are limited to grabbing the objects.

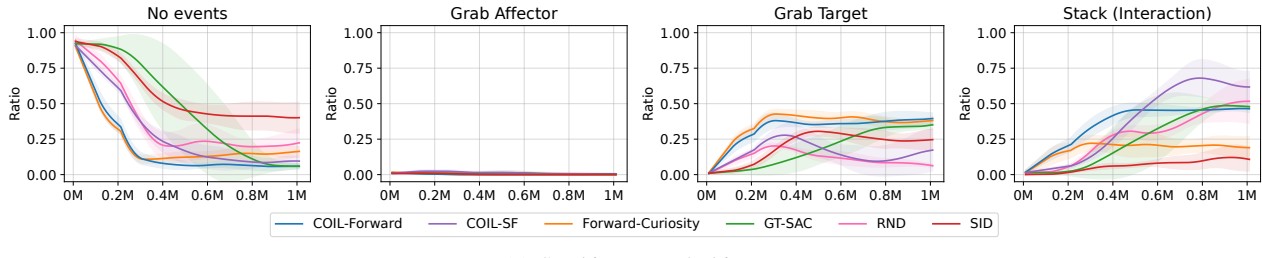

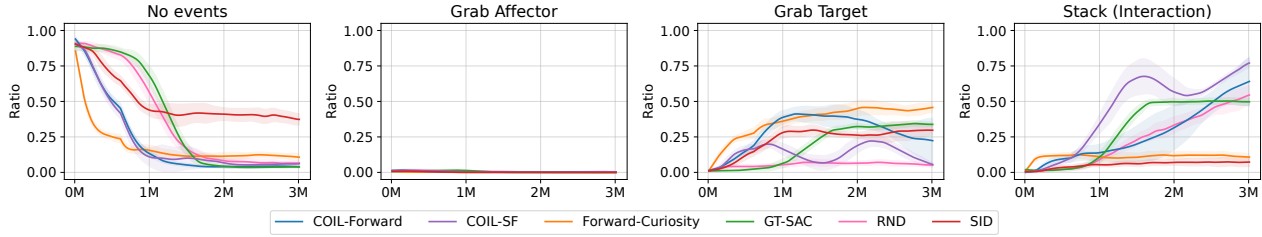

Figure 6: The ratios of the states visited during the episodes for each label in Stacking Box. (1) No events: States without any grabbing or stacking event, (2) Grab Affector: States in which the agent grabs the affector, (3) Grab Target: States in which the agent grabs the target, (4) Stack: States in which the target is stacked on the affector.

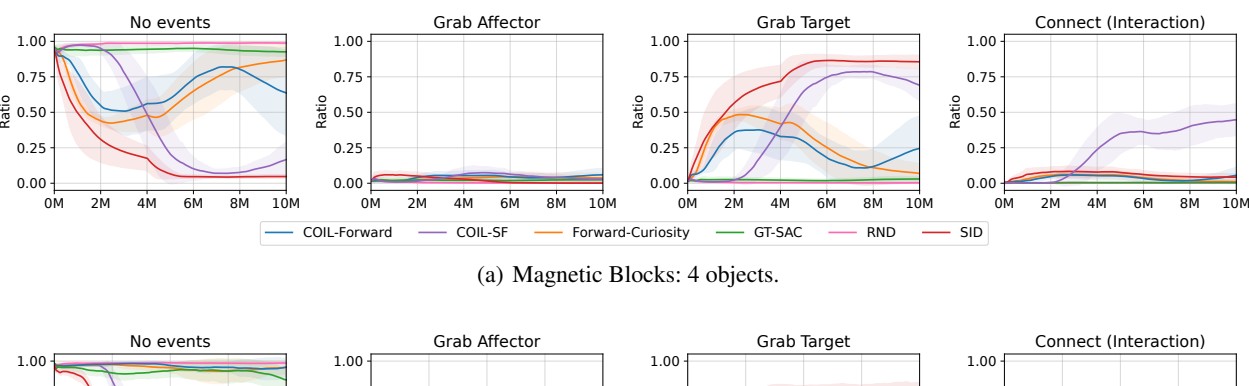

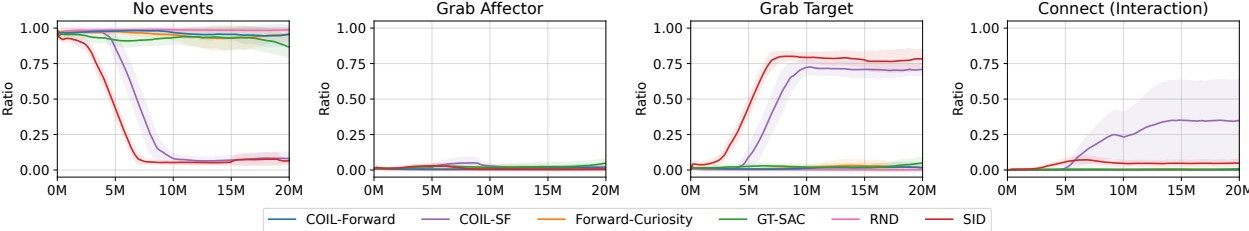

Figure 7: The ratios of the states visited during the episodes for each label in Magnetic Blocks. (1) No events: States without any grabbing or connecting event, (2) Grab Affector: States in which the agent grabs the affector, (3) Grab Target: States in which the agent grabs the target, (4) Connect: States in which the target and the affector are connected.

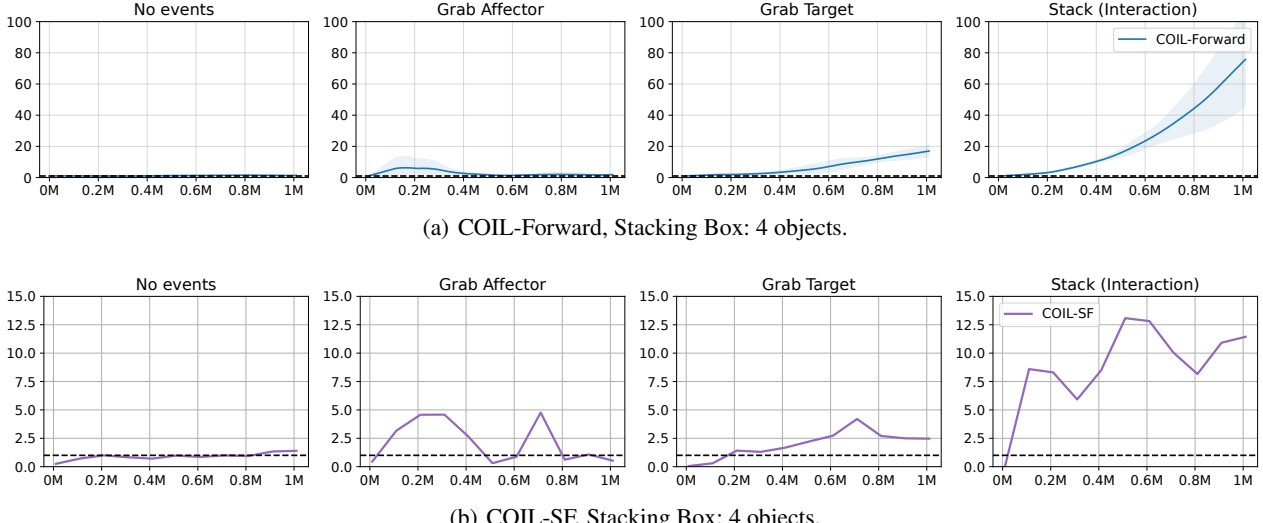

(a) COIL-Forward, Stacking Box: 4 objects.

(b) COIL-SF, Stacking Box: 4 objects.

Figure 8: The ratio of counterfactual prediction error to epistemic uncertainty of dynamics models for each label in COIL, Stacking Box. (1) No events: States without any grabbing or stacking event, (2) Grab Affector: States in which the agent grabs the affector, (3) Grab Target: States in which the agent grabs the target, (4) Stack: States in which the target is stacked on the affector. Higher ratio means that a higher reward $r_{\text{COIL}}$ will be given.

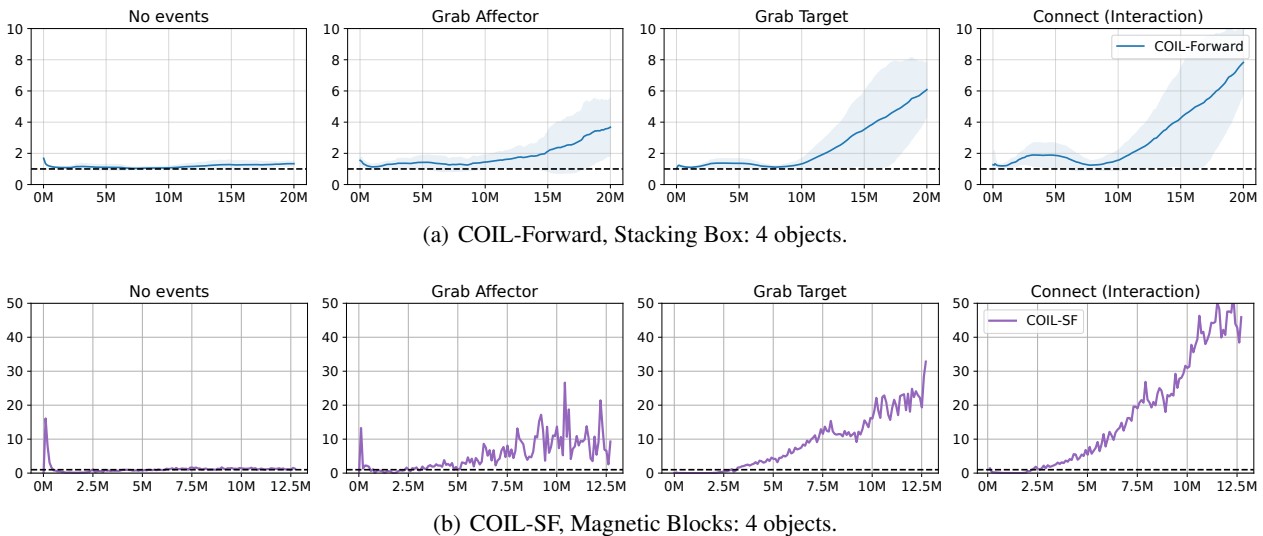

(a) COIL-Forward, Stacking Box: 4 objects.

(b) COIL-SF, Magnetic Blocks: 4 objects.

Figure 9: The ratio of counterfactual prediction error to epistemic uncertainty of dynamics models for each label in COIL, Magnetic Blocks. (1) No events: States without any grabbing or connecting event, (2) Grab Affector: States in which the agent grabs the affector, (3) Grab Target: States in which the agent grabs the target, (4) Connect: States in which the target and the affector are connected. Higher ratio means that a higher reward $r_{\text{COIL}}$ will be given.

