# OpenReview forum: "Unsupervised Object Interaction Learning with Counterfactual Dynamics Models"
_ICLR.cc/2023/Workshop/RRL — RRL 2023 Poster_

### Official Review · Reviewer_WzEt · 2023-02-27
**A clear and well-motivated approach to object-oriented exploration**

**Rating:** 4
**Confidence:** 4

**Review:**

# Summary
The author present a novel algorithm for encouraging object interactions in an unsupervised exploration setting.
They propose a novel intrinsic reward formulation based on counterfactual reasoning, and show that this method outperforms baselines on multi-objects environments.

# Significance, Novelty, Impact
The paper proposes a novel intrinsic reward formulation, with two variants -- namely, forward model and successor features -- which are well-defined and motivated.
The reward formulation is general and has a great potential in tasks that contain multiple objects, for which previous intrinsic reward formulation could converge to a low hanging fruit behaviour.

# Empirical Evaluation
The method is validated empirically in two different object manipulation tasks.
The experiments contains comparison against unsupervised exploration baselines, as well as a topline with domain knowledge.
The method outperforms baselines on all tasks, and is the only method succeeding in performing object interaction in the harder task.
The author also presented extensive analysis to understand the behaviour of both variants of their method, as well as generalization experiments to unseen objects.

# Writing
The paper is clear and well-written, and succeeds in providing an intuition of both problem and method, and the balance is good between formal description and intuitive explanation.
It provides extensive evaluation, analysis and qualitative visualisation of the method.

# Remark
One interesting ablation would be to compare several types of interventions (random perturbations vs. more object-specific interventions), and study the impact on the downstream task performance.

---

### Official Review · Reviewer_W8iv · 2023-02-28
**Relevant and interesting for RRL**

**Rating:** 4
**Confidence:** 3

**Review:**

**Summary**

This work considers the problem of learning skills that are useful for acting on multiple objects. They propose an intrinsic reward that is derived from counterfactual reasoning: if an agent observes a transition where two objects interact with each other (e.g., by stacking them), how does this transition differ from an imagined/counterfactual world where one of the objects has been displaced or perturbed such that no interaction occurs? The difference between the observed transitions and counterfactual transitions (estimated with either a learned forward model or successor features) becomes an intrinsic reward signal (where more difference is better). The authors compare their intrinsic reward with several others from the literature and show that their method leads to a higher success rate in producing object-object interactions in environments without extrinsic reward: either stacking boxes on top of each other or magnetically connecting two objects together.

**Relevance and significance**

This paper is very relevant and interesting to RRL because it tackles skill learning. The kind of object-object interaction skill developed by this work can be relevant for accelerating learning in downstream tasks that require some composition of objects, e.g., stacking blocks into a certain configuration or, to give an example in the real world, combining furniture components into a whole (example given by authors).

**Quality and clarity**

The paper is well-written and clear.

**Feedback**

I would be interested in seeing an application/fine-tuning of this work's learned skills to a downstream task (where there is an extrinsic reward), even something simple like rewarding the agent for stacking exactly 3 blocks. The reward accumulated in a downstream task would be able to measure how useful the learned skill is. In contrast, the current measure of success rate seems to be entangled with the proposed intrinsic reward and only seems to measure whether the intrinsic reward is working as expected/desired.